# CDDO-Me Abrogates Aberrant Mitochondrial Elongation in Clasmatodendritic Degeneration by Regulating NF-κB-PDI-Mediated S-Nitrosylation of DRP1

**DOI:** 10.3390/ijms24065875

**Published:** 2023-03-20

**Authors:** Duk-Shin Lee, Tae-Hyun Kim, Hana Park, Ji-Eun Kim

**Affiliations:** 1Department of Anatomy and Neurobiology, College of Medicine, Hallym University, Chuncheon 24252, Republic of Korea; 2Institute of Epilepsy Research, College of Medicine, Hallym University, Chuncheon 24252, Republic of Korea

**Keywords:** astrocyte, autophagy, CDDO-Me, mitochondrial dynamics, S-nitrosylation, seizure, SN50

## Abstract

Clasmatodendrosis is a kind of astroglial degeneration pattern which facilitates excessive autophagy. Although abnormal mitochondrial elongation is relevant to this astroglial degeneration, the underlying mechanisms of aberrant mitochondrial dynamics are still incompletely understood. Protein disulfide isomerase (PDI) is an oxidoreductase in the endoplasmic reticulum (ER). Since PDI expression is downregulated in clasmatodendritic astrocytes, PDI may be involved in aberrant mitochondrial elongation in clasmatodendritic astrocytes. In the present study, 26% of CA1 astrocytes showed clasmatodendritic degeneration in chronic epilepsy rats. 2-cyano-3,12-dioxo-oleana-1,9(11)-dien-28-oic acid methyl ester (CDDO-Me; bardoxolone methyl or RTA 402) and SN50 (a nuclear factor-κB (NF-κB) inhibitor) ameliorated the fraction of clasmatodendritic astrocytes to 6.8 and 8.1% in CA1 astrocytes, accompanied by the decreases in lysosomal-associated membrane protein 1 (LAMP1) expression and microtubule-associated protein 1A/1B light-chain 3 (LC3)-II/LC3-I ratio, indicating the reduced autophagy flux. Furthermore, CDDO-Me and SN50 reduced NF-κB S529 fluorescent intensity to 0.6- and 0.57-fold of vehicle-treated animal level, respectively. CDDO-Me and SN50 facilitated mitochondrial fission in CA1 astrocytes, independent of dynamin-related protein 1 (DRP1) S616 phosphorylation. In chronic epilepsy rats, total PDI protein, S-nitrosylated PDI (SNO-PDI), and SNO-DRP1 levels were 0.35-, 0.34- and 0.45-fold of control level, respectively, in the CA1 region and increased CDDO-Me and SN50. Furthermore, PDI knockdown resulted in mitochondrial elongation in intact CA1 astrocytes under physiological condition, while it did not evoke clasmatodendrosis. Therefore, our findings suggest that NF-κB-mediated PDI inhibition may play an important role in clasmatodendrosis via aberrant mitochondrial elongation.

## 1. Introduction

In the brain, astrocytes maintain a state of balance in the acid–base equilibrium, energy metabolism, and neuronal activity [1]. Therefore, the dysfunction of astrocytes is one of the causes of neurological diseases, including epilepsy [2,3]. Vacuolized degeneration represents an irreversible fatal change in astrocytes [4], which was first reported by Alzheimer in 1910 and termed “clasmatodendrosis” by Cajal [5]. Clasmatodendrosis is evoked by failure of bioenergetics and acidosis coupled to impaired mitochondrial functions under pathophysiological conditions, such as Alzheimer disease, brain ischemia, and epileptic seizures [3,6,7,8,9,10,11]. In previous studies, we have reported that clasmatodendritic astrocytes show the upregulations in lysosomal-associated membrane protein 1 (LAMP1), beclin-1, and microtubule-associated protein 1A/1B light-chain 3 (LC3) expression, which indicates the activation of autophagic processes [12,13]. Basically, autophagy is the process to remove aberrant components for cell survival [14]. However, excessive autophagy evokes type II programmed cell death, known as autophagic cell death [15]. Indeed, clasmatodendritic astrocytes include condensed chromatin, lysosomes, and large membrane-bound osmiophilic cytoplasmic inclusions [16,17]. Therefore, we have reported for the first time that clasmatodendrosis is one of the astroglial degenerations which is relevant to excessive autophagy [12,13].

The clasmatodendritic process is regulated by several signaling molecules, including nuclear factor-erythroid 2-related factor 2 (Nrf2), extracellular signal-related kinases 1/2 (ERK1/2), nuclear factor-κB (NF-κB), 5′ adenosine monophosphate-activated protein kinase (AMPK), and AKT-mediated pathways [10,11,18]. However, impaired mitochondrial dynamics are also involved in clasmatodendrosis. Briefly, clasmatodendritic astrocytes show aberrant mitochondrial elongation induced by the increased AKT-mediated dynamin-related protein 1 (DRP1; a mitochondrial fission protein) serine (S) 637 phosphorylation [19,20]. Thus, it is noteworthy to elucidate the relationships between clasmatodendrosis-related signaling pathways and mitochondrial dynamics, which are still incompletely understood.

Protein disulfide isomerase (PDI) is an oxidoreductase regulating protein folding in the endoplasmic reticulum (ER) [21]. Interestingly, PDI plays the role of a nitric oxide (NO) donor to DRP1 and exerts its S616 phosphorylation, which facilitates mitochondrial fission (fragmentation) in CA1 neurons [22]. Furthermore, PDI expression is downregulated in clasmatodendritic astrocytes compared to intact astrocytes [23]. Therefore, it is likely that downregulation of PDI may be involved in aberrant mitochondrial elongation in clasmatodendritic astrocytes.

2-Cyano-3,12-dioxo-oleana-1,9(11)-dien-28-oic acid methyl ester (CDDO-Me; bardoxolone methyl or RTA 402) is an Nrf2 activator that affects mitochondrial dynamics/mitophagy in various neurological diseases [24,25]. Indeed, CDDO-Me attenuates clasmatodendrosis and abnormal mitochondrial elongation in CA1 astrocytes by reducing AKT activity [11,22]. Furthermore, CDDO-Me directly inhibits the NF-κB signaling pathway [26,27]. Regarding these previous reports, it is plausible that CDDO-Me may also attenuate clasmatodendrosis by inhibiting NF-κB signaling pathway and/or recovering the impairment of PDI/DRP1-mediated mitochondrial fission. Therefore, the present study was conducted to further clarify the underlying mechanisms of impaired mitochondrial dynamics during clasmatodendrosis.

Here, we demonstrate that CDDO-Me and SN50 (a NF-κB inhibitor) ameliorated clasmatodendrosis in CA1 astrocytes in the hippocampus of chronic epilepsy rats accompanied by the reduced NF-κB S529 phosphorylation. CDDO-Me and SN50 facilitated mitochondrial fission in CA1 astrocytes, independent of DRP1 S616 phosphorylation. Both CDDO-Me and SN50 increased total PDI protein and S-nitrosylated (SNO)-PDI levels in CA1 astrocytes of chronic epilepsy rats. PDI knockdown resulted in mitochondrial elongation in intact CA1 astrocytes under physiological conditions, while it did not evoke clasmatodendrosis. Therefore, our findings suggest that NF-κB-mediated PDI inhibition may regulate clasmatodendritic degeneration via aberrant mitochondrial elongation.

## 2. Results

### 2.1. CDDO-Me and SN50 Ameliorates Clasmatodendritic Degeneration in CA1 Astrocytes

In total, 26% of CA1 astrocytes showed clasmatodendritic degeneration in chronic epilepsy rats (*t*_(12)_ = 9.3, *p* < 0.001 vs. control rats, Student *t*-test, *n* = 7; Figure 1A,B), accompanied by LAMP1 upregulation (*t*_(12)_ = 9.5, *p* < 0.001 vs. control rats, Student *t*-test, *n* = 7; Figure 1A,C) and LC3 induction (*t*_(12)_ = 12.1, *p* < 0.001 vs. control rats, Student *t*-test, *n* = 7; Figure 1A,D). CDDO-Me reduced the fraction of clasmatodendritic astrocytes to 6.8 % in CA1 astrocytes (*F*_(3,24)_ = 52.1, *p* < 0.001, one-way ANOVA, *n* = 7; Figure 1A,B) and LAMP1 fluorescent intensity to 0.75-fold of vehicle-treated animal level (*F*_(3,24)_ = 36.7, *p* < 0.001, one-way ANOVA, *n* = 7; Figure 1A,C). In addition, CDDO-Me decreased LC3 fluorescent intensity to 0.64-fold of vehicle-treated animal levels (*F*_(3,24)_ = 52.3, *p* < 0.001, one-way ANOVA, *n* = 7; Figure 1A,D). Thus, some vacuolized CA1 astrocytes did not contain LC3 expression (Figure 1A). Similar to CDDO-Me, SN50 also decreased the fraction of clasmatodendritic astrocytes to 8.1% in CA1 astrocytes (*F*_(3,24)_ = 52.1, *p* < 0.001, one-way ANOVA, *n* = 7; Figure 1A,B) and LAMP1 fluorescent intensity to 0.74-fold of vehicle-treated animal levels (*F*_(3,24)_ = 36.7, *p* < 0.001, one-way ANOVA, *n* = 7; Figure 1A,C). SN50 also decreased LC3 fluorescent intensity to 0.62-fold of vehicle-treated animal level (*F*_(3,24)_ = 52.3, *p* < 0.001, one-way ANOVA, *n* = 7; Figure 1A,D). 

On the other hand, the punctate distribution of LC3 indicates its concentration on the autophagosomes [14]. On high magnification images, in the present study, numerous LC3-positive puncta were apparently observed in vacuolized astrocytes of vehicle-treated epilepsy rats (Figure 2A). Both CDDO-Me and SN50 diminished the number of LC3-positive puncta in clasmatodendritic astrocytes (Figure 2A). Furthermore, Western blot data revealed that LAMP1 (*F*_(3,24)_ = 34.4, *p* < 0.001, one-way ANOVA, *n* = 7), LC3-I (*F*_(3,24)_ = 11.6, *p* < 0.001, one-way ANOVA, *n* = 7), and LC3-II (*F*_(3,24)_ = 77.5, *p* < 0.001, one-way ANOVA, *n* = 7) densities were upregulated in the epileptic hippocampus, which were effectively decreased by CDDO-Me and SN50 (Figure 2B–E and Appendix A). In addition, the LC3-II/LC-I ratio was increased in the epileptic hippocampus, which was also diminished by CDDO-Me and SN50 (*F*_(3,24)_ = 18, *p* < 0.001, one-way ANOVA, *n* = 7; Figure 2F). Thus, our findings indicate that autophagy flux may be increased in clasmatodendritic astrocytes which were attenuated by CDDO-Me and SN50. 

Considering the inhibitory effects of CDDO-Me and SN50 on NF-κB activity [26,27], these findings indicate that the NF-κB-mediated signaling pathway may regulate clasmatodendritic CA1 astroglial degeneration in the epileptic hippocampus. To confirm this, we evaluated the efficacies of CDDO-Me and SN50 in NF-κB S529 phosphorylation which results in clasmatodendritic degeneration [13].

### 2.2. CDDO-Me and SN50 Inhibit NF-κB S529 Phosphorylation in CA1 Astrocytes

Consistent with our previous study [13], NF-κB S529 signals were localized in dissociated nuclei of clasmatodendritic CA1 astrocytes (Figure 3A). In addition, NF-κB S529 phosphorylation was increased to 2.1-fold in clasmatodendritic CA1 astrocytes compared to control rats (*t*_(12)_ = 8.4, *p* < 0.001, Student *t*-test, *n* = 7; Figure 3A,B). CDDO-Me reduced NF-κB S529 fluorescent intensity to 0.6-fold of vehicle-treated animal level (*F*_(3,24)_ = 48.8, *p* < 0.001, one-way ANOVA, *n* = 7; Figure 3A,B). SN50 also diminished NF-κB S529 fluorescent intensity to 0.57-fold of vehicle-treated animal level (*F*_(3,24)_ = 48.8, *p* < 0.001, one-way ANOVA, *n* = 7; Figure 3A,B). Thus, our findings indicate that the intensified nuclear NF-κB S529 phosphorylation may lead to clasmatodendrosis in CA1 astrocytes.

### 2.3. CDDO-Me and SN50 Ameliorate Aberrant Mitochondrial Elongation in CA1 Astrocytes

Since abnormal mitochondrial hyperfusion results in clasmatodendrosis [19], we validated the effects of CDDO-Me and SN50 on mitochondrial fusion in CA1 astrocytes. In control animals, the area-weighted form factor (indicative of mitochondrial elongation [28,29]) was 2.48 (Figure 4A,B). In chronic epilepsy rats, mitochondrial elongation was 6.11 (*t*_(68)_ = 6.4, *p* < 0.001, Student *t*-test, *n* = 7; Figure 4A,B). CDDO-Me decreased it to 3.98 (*F*_(2,102)_ = 6.7, *p* = 0.002, one-way ANOVA, *n* = 7; Figure 4A,B). SN50 also reduced mitochondrial elongation to 3.29 (*F*_(2, 102)_ = 7.2, *p* = 0.001, one-way ANOVA, *n* = 7; Figure 4A,B). The cumulative area:perimeter ratio (indicative of the mitochondrial fusion or aggregation [28,29]) was 2.48 in control rats. In chronic epilepsy rats, the cumulative area:perimeter ratio was 13.11 (*t*_(68)_ = 9.4, *p* < 0.001, Student *t*-test, *n* = 7). CDDO-Me diminished it to 5.63 (*F*_(2, 102)_ = 11.8, *p* < 0.001, one-way ANOVA, *n* = 7; Figure 4C). SN50 reduced the cumulative area:perimeter ratio to 4.89 (*F*_(2, 102)_ = 13.1, *p* < 0.001, one-way ANOVA, *n* = 7; Figure 4C). In control animals, the form factor (indicative of transition of individual mitochondrion to complex shape [28,29]) was 2.11. The form factor was reduced to 1.31 in chronic epilepsy rats (*t*_(68)_ = 4.8, *p* < 0.001, Student *t*-test, *n* = 7; Figure 4C). CDDO-Me elevated it to 3.29 (*F*_(2, 102)_ = 9.8, *p* < 0.001, one-way ANOVA, *n* = 7; Figure 4C). SN50 also increased the form factor to 2.97 (*F*_(2, 102)_ = 10.3, *p* < 0.001, one-way ANOVA, *n* = 7; Figure 4C). These findings indicate that CDDO-Me and SN50 may attenuated the accumulation of hyperfused mitochondria in CA1 astrocytes by inhibiting NF-κB S529 phosphorylation.

### 2.4. CDDO-Me and SN50 Induce Mitochondrial Fragmentation without Altering DRP1 S616 Phosphorylation

DRP1 is one of the regulators for mitochondrial dynamics whose activity is oppositely modulated by two distinct sites: phosphorylation of DRP1 at S616 promotes mitochondrial fragmentation, but S637 phosphorylation inhibits mitochondrial fission [25,30,31,32]. Recently, we have reported that CDDO-Me decreases astroglial DRP1 S637 phosphorylation level without affecting S616 phosphorylation [20]. Since CDDO-Me and SN50 led to mitochondrial fragmentation in the present study, we confirmed that their effects on DRP1 S616 phosphorylation in CA1 astrocytes. In chronic epilepsy rats, the DRP1 S616 fluorescent intensity was 0.64-fold of control level in clasmatodendritic CA1 astrocytes (*t*_(12)_ = 8.2, *p* < 0.001 vs. control animals, Student *t*-test, *n* = 7; Figure 5A,B). Neither CDDO-Me nor SN50 affected DRP1 S616 fluorescent intensity in CA1 astrocytes (*F*_(2,18)_ = 0.4, *p* = 0.696, one-way ANOVA, *n* = 7; Figure 5A,B). Therefore, it is likely that CDDO-Me and SN50 may attenuate aberrant mitochondrial elongation in CA1 astrocytes, independent of DRP1 S616 phosphorylation.

### 2.5. CDDO-Me and SN50 Restore the Reduced PDI Expression in CA1 Astrocytes

PDI is a chaperone in ER and also presents in cytoplasm and cell surface [21]. Although PDI acts as thiol-disulfide exchanger, it also plays a role as a transporter of NO residue [33]. In particular, we have reported that PDI regulates S-nitrosylation of DRP1 and accelerates mitochondrial fission [22] since S-nitrosylation induces DRP1 dimerization, which directly increases its activity [34]. Considering that PDI expression is downregulated in clasmatodendritic astrocytes compared to intact astrocytes [23], it is likely that CDDO-Me and SN50 may restore the downregulated PDI expression and recover aberrant mitochondrial elongation in clasmatodendritic astrocytes.

Consistent with our previous study [23], PDI fluorescent intensity was 0.34-fold of control level in clasmatodendritic CA1 astrocytes (*t*_(12)_ = 21.1, *p* < 0.001 vs. control animals, Student *t*-test, *n* = 7; Figure 6A,B). CDDO-Me increased PDI fluorescent intensity to 0.69-fold of control level in CA1 astrocytes (*F*_(3,24)_ = 104.5, *p* < 0.001, one-way ANOVA, *n* = 7; Figure 6A,B). SN50 also enhanced it to 0.72-fold of control level in CA1 astrocytes (*F*_(3,24)_ = 104.5, *p* < 0.001, one-way ANOVA, *n* = 7; Figure 6A,B). Thus, it is likely that CDDO-Me and SN50 would inhibit mitochondrial hyperfusion by enhancing PDI-mediated S-nitrosylation of DRP1. To confirm this hypothesis, we evaluated the effects of CDDO-Me and SN50 on mitochondrial dynamics in the epileptic hippocampus.

Western blot revealed that total PDI protein and S-nitrosylated PDI (SNO-PDI) levels were 0.35- (*t*_(12)_ = 22.3, *p* < 0.001 vs. control animals, Student *t*-test, *n* = 7; Figure 7A,B and Appendix A) and 0.34-fold (*t*_(12)_ = 21.9, *p* < 0.001 vs. control animals, Student *t*-test, *n* = 7; Figure 7A,C) of control level in the CA1 region. CDDO-Me increased total PDI level to 0.67-fold of control level in the CA1 region (*F*_(2,18)_ = 44.2, *p* < 0.001, one-way ANOVA, *n* = 7; Figure 7A,B). SN50 also upregulated total PDI level to 0.7-fold of control level in the CA1 region (*F*_(2,18)_ = 44.2, *p* < 0.001, one-way ANOVA, *n* = 7; Figure 7A,B). Compatible with total PDI level, CDDO-Me enhanced SNO-PDI level to 0.68-fold of control level in the CA1 region (*F*_(2,18)_ = 53.3, *p* < 0.001, one-way ANOVA, *n* = 7; Figure 7A,C). SN50 also increased it to 0.7-fold of control level in the CA1 region (*F*_(2,18)_ = 53.3, *p* < 0.001, one-way ANOVA, *n* = 7; Figure 7A,C). Although CDDO-Me and SN50 did not affect the reduced total DRP1 level (*F*_(2,18)_ = 0.1, *p* = 0.927, one-way ANOVA, *n* = 7; Figure 7A,D), CDDO-Me enhanced SNO-DRP1 level to 0.72-fold of control level in the CA1 region (*F*_(2,18)_ = 24.6, *p* < 0.001, one-way ANOVA, *n* = 7; Figure 7A,E). SN50 increased it to 0.7-fold of control level in the CA1 region (*F*_(2,18)_ = 24.6, *p* < 0.001, one-way ANOVA, *n* = 7; Figure 7A,E). These findings indicate that the elevated NF-κB S529 phosphorylation may downregulate PDI protein expression, inhibit PDI-mediated S-nitrosylation of DRP1 and subsequently lead to aberrant mitochondrial elongation in clasmatodendritic CA1 astrocytes, which would be mitigated by CDDO-Me and SN50.

### 2.6. PDI Knockdown Leads to Mitochondrial Elongation in CA1 Astrocytes under Physiological Condition

To elucidate whether the downregulated PDI protein level abrogates DRP1-medaited mitochondrial fission in CA1 astrocytes, we applied PDI siRNA to normal rats. Compared to control siRNA, PDI siRNA decreased PDI protein level in the CA1 region (*t*_(12)_ = 15.2, *p* < 0.001 vs. control siRNA, Student *t*-test, *n* = 7; Figure 8A,B and Appendix A) without affecting DRP1 protein level (Figure 8A,C). PDI knockdown also reduced SNO-DRP1 level (*t*_(12)_ = 10.9, *p* < 0.001 vs. control siRNA, Student *t*-test, *n* = 7; Figure 8A,D). Although PDI siRNA did not evoke clasmatodendritic degeneration, it increased mitochondrial elongation (*t*_(68)_ = 4.1, *p* < 0.001, Student *t*-test, *n* = 7; Figure 8E,F), the cumulative area:perimeter ratio (*t*_(68)_ = 7.9, *p* < 0.001, Student *t*-test, *n* = 7; Figure 8G), and the form factor (*t*_(68)_ = 4.7, *p* < 0.001, Student *t*-test, *n* = 7; Figure 8G). These findings indicate that the reduced PDI-mediated S-nitrosylation of DRP1 may be involved in the impairment of mitochondrial fission in CA1 astrocytes during clasmatodendritic process.

## 3. Discussion

The major findings in the present study are that CDDO-Me and SN50 attenuated clasmatodendrosis in CA1 astrocytes by restoring PDI-mediated mitochondrial fission, suggesting that NF-κB-mediated PDI downregulation may lead to clasmatodendrosis in CA1 astrocytes (Figure 9).

Clasmatodendrosis shows autophagic phenomena in response to epileptic seizures through NF-κB S529 phosphorylation [12,13], which is closely relevant to the synchronous epileptiform discharges [3,11]. Under stressful condition, the cell organelles enveloped by double membrane vesicles (autophagosomes) and delivered to the lysosomes for digestion and the subsequent recycle of amino acid into the cell machinery [35]. However, dysregulation of this autophagic process leads to non-apoptotic programmed cell death [14,36]. Although NF-κB phosphorylation is involved in the regulation of autophagic process [37], little data are available to explain the downstream effectors of NF-κB-mediated clasmatodendrosis.

In the present study, we found that CDDO-Me and SN50 abrogated clasmatodendrosis in CA1 astrocytes by inhibiting NF-κB S529 phosphorylation. CDDO-Me directly or indirectly inhibits the NF-κB pathway [38]. Furthermore, CDDO-Me inhibits casein kinase 2 (CK2) that regulates NF-κB activity [11,39]. Since CK2 enhances NF-κB nuclear transcriptional activity by S529 phosphorylation [40,41], our findings indicate that CDDO-Me may also ameliorate clasmatodendrosis by inhibiting CK2-mediated NF-κB S529 phosphorylation.

PDI is responsible for modulating disulfide bond formation [42]. Under ER stress, PDI removes misfolded proteins to maintain the ER homeostasis [43,44]. Thus, upregulation of PDI expression is an adaptive response to protect cells from ER stress [45,46]. However, PDI also induces mitochondrial membrane permeabilization leading to apoptosis [47]. Consistent with a previous study [23], the present data demonstrate that PDI expression was downregulated in clasmatodendritic CA1 astrocytes. Since upregulated PDI expression results in acquisition of tolerance against detrimental stress in astrocytes [46] and excessive unfolded protein response (UPR) induced by accumulating misfolding and aggregation of proteins leads to apoptosis or autophagy [48,49], it is not surprising that PDI downregulation may be involved in clasmatodendrosis in CA1 astrocytes. In the present study, interestingly, CDDO-Me and SN50 increased PDI expression in CA1 astrocytes, concomitant with the abrogation of NF-κB S529 phosphorylation. These findings indicate that the NF-κB S529 phosphorylation may trigger clasmatodendrosis by reducing PDI expression.

On the other hand, PDI deletion mitigates neuroinflammation after traumatic brain injury (TBI) in mice, which is associated with the decreased NF-κB phosphorylation [50]. However, the roles of PDI in NF-κB activity are still controversial. PDI ablation inactivates NF-κB phosphorylation [51]. In contrast, over-expression of PDI suppresses NF-κB activity [52]. Furthermore, tumor necrosis factor-α (TNF-α)-stimulated NF-κB signaling is unaffected by PDI knockdown [21]. Considering that the TNF-α neutralization attenuates clasmatodendritic astrocytes accompanied by reduced p65/RelA-Ser529 phosphorylation [13], it is likely that NF-κB may be an upstream regulator for PDI rather than its downstream effector for activity at least during clasmatodendritic process.

The one of the underlying molecular mechanisms of clasmatodendrosis is mitochondrial defects initiated by acidosis [8,53,54,55]. Furthermore, aberrant mitochondrial elongation is one of causes of clasmatodendrosis [19,20]. The morphology of mitochondria is altered by cellular energetic status. Mitochondrial fusion maintains and restore mitochondrial function by facilitating the redistribution of mitochondrial components. Mitochondrial fission is required for the mitophagy to remove damaged/defective mitochondria via mitophagy [56,57,58,59]. Therefore, imbalance in mitochondrial dynamics lead to cell death [60,61]. Mitochondrial dynamics are regulated by various molecules including DRP1. DRP1 is a regulator of mitochondrial fission. Phosphorylation of DRP1 at S616 promotes mitochondrial fission, while S637 phosphorylation inhibits mitochondrial fission [58,59,60]. Therefore, the balance between DRP1 S616 and S637 phosphorylation is tightly regulated for cell viability [25,30,31,32]. Indeed, the decreased DRP1-S616/S637 phosphorylation ratio results in abnormal mitochondrial hyperfusion in clasmatodendritic CA1 astrocytes without altering other mitochondrial dynamics-related molecules [19,20]. In the present study, CDDO-Me and SN50 ameliorated aberrant mitochondrial hyperfusion in CA1 astrocytes without affecting DRP1 S616 phosphorylation. Therefore, our findings indicate that CDDO-Me and SN50 may attenuate clasmatodendrosis in CA1 astrocytes by recovering DRP1-mediated mitochondrial fragmentation, independent of DRP1 S616 phosphorylation.

In addition to the phosphorylation, S-nitrosylation of the cysteine (C) 644 site enhances DRP1 GTPase activity and its oligomerization in association with excessive mitochondrial fission [34]. S-nitrosylation is a post-translational modification induced by the covalent binding of NO to a cysteine thiol group of the protein. This modification affects a variety of cellular processes, protein function and protein-protein interactions [62]. Although S-nitrosylation inhibits PDI function [63,64], PDI acts as NO donor of DRP1 to exert S616 phosphorylation, which facilitates mitochondrial fission (fragmentation) [22]. In the present study, PDI expression is downregulated in clasmatodendritic astrocytes compared to intact astrocytes. Furthermore, CDDO-Me and SN50 restored the decreased PDI expression in clasmatodendritic astrocytes and increased SNO-PDI level in the CA1 region without total DRP1 protein level. Therefore, our findings indicate that the reduction in PDI-mediated S-nitrosylation of DRP1 may result in aberrant mitochondrial elongation in CA1 astrocytes during clasmatodendritic process. Indeed, PDI knockdown led to mitochondrial elongation in CA1 astrocytes under physiological condition. Taken together, our findings suggest that NF-κB S529 phosphorylation may decrease PDI expression, which would abrogate appropriate mitochondrial fragmentation by reducing S-nitrosylation of DRP1 and consequently evoke clasmatodendrosis.

Recently, we have reported that various enzymes regulate clasmatodendrosis. In particular, AKT, glutathione peroxidase 1 (GPx1), and peroxiredoxin 6 (Prdx6) play important roles in clasmatodendritic degeneration [11,20,65,66,67]. The dysregulation of the Prdx6-GPx1-mediated signaling pathway exerts clasmatodendrosis by augmenting oxidative stress [65,66,67]. Oxidative stress also induces AKT activation that phosphorylates DRP1 S637 and facilitates bax-interacting factor 1 (Bif-1)-mediated autophagy [11,20]. In the present study, we found that both CDDO-Me and SN50 ameliorated aberrant mitochondrial hyperfusion by recovering NF-κB-PDI-mediated S-nitrosylation of DRP1. Interestingly, CK2 is an upstream regulator of both NF-κB S529 phosphorylation and AKT activation during clasmatodendrosis [11,39,40,41]. Considering oxidative stress-induced CK2 activation [11] and the anti-oxidant properties of CDDO-Me [24,25], our findings suggest that oxidative stress may cause CK2 hyperactivation eliciting clasmatodendrosis through the NF-κB-PDI- and AKT-Bif-1-mediated signaling pathways.

In previous studies, we reported that clasmatodendrosis is Tdt-mediated dUTP Nick-End Labeling (TUNEL)-negative coagulative necrosis in astrocytes [3,9]. Later, we found that vacuoles in clasmatodendritic astrocytes contain LC3 and LAMP1 signals. Since LC3 is required for the autophagosome formation and LAMP1 is a marker for lysosomal biogenesis, amounts, and morphology [68,69,70,71], we reported that large vacuoles in clasmatodendritic astrocytes are autolysosomes [12,13]. LAMP1 is the predominant lysosomal membrane protein to maintain the integrity of the lysosomal membrane and the clearance of autophagosomes [68,69]. Cytoplasmic form LC3 (LC3-I) is diffusely observed in cell bodies, which is modified to LC3-II and concentrated in autophagosomes that exhibits granular puncta and a different mobility in electrophoresis [14,37,70,71]. Thus, an increase in LC3-II/LC3-I ratio is indicative of activation of autophagy process [37,70,71]. In the present study, clasmatodendritic CA1 astrocytes showed the increased LAMP1 and LC3 expression, which were ameliorated by CDDO-Me and SN50. Furthermore, LC3-positive puncta were apparently detected in vacuolized astrocytes of vehicle-treated epilepsy rats. Western blot data also revealed that LAMP1, LC3-I and LC3-II densities were elevated in the epileptic hippocampus concomitant with the increased LC3-II/LC3-I ratio, which were attenuated by CDDO-Me and SN50. Compatible with the present data, Sakai et al. [70] report that clasmatodendrosis is relevant to UPS-mediated autophagy. Qin et al. [71] also demonstrate that ischemia-injured astrocytes contain numerous multimembrane vesicles, described as typical for autophagosomes, which eventually fused with lysosomes in the cytoplasm, indicating that the autophagic/lysosomal pathway activation contributes to the decreased viability of astrocytes. Therefore, our findings suggest that excessive autophagy is involved in the pathogenesis of clasmatodendritic degeneration in the epileptic hippocampus, although it is unclear whether aberrant autophagy is the main cause of clasmatodendrosis or directly leads to astroglial degeneration.

## 4. Materials and Methods

### 4.1. Experimental Animals and Chemicals

Male Sprague–Dawley (SD) rats (7 weeks old) were used in the present study. Animals were cared under standard condition (22 ± 2 °C, 55 ± 5% and a 12:12 light/dark cycle with lights). Animals were provided with a commercial diet and water ad libitum. Experimental protocols were approved by the Institutional Animal Care and Use Committee of Hallym University (Chuncheon, South Korea, Code number: Hallym 2018-3, approval date: 30 April 2018 and Hallym 2021-30, approval date: 17 May 2021). All reagents were obtained from Sigma-Aldrich (St. Louis, MO, USA), except as noted.

### 4.2. Generation of Chronic Epilepsy Rats

Rats were given LiCl (127 mg/kg, i.p.) 1 day before the pilocarpine administration (30 mg/kg, i.p.). Twenty minutes before pilocarpine treatment, rats were treated with atropine methylbromide (5 mg/kg i.p.). Two hours after status epilepticus (SE) on-set, animals were given with diazepam (Valium; Hoffman la Roche, Neuilly sur-Seine, France; 10 mg/kg, i.p.) and repeated, as needed. Control animals received saline. SE-experienced rats were video-monitored 8 h a day to classify chronic epilepsy rats.

### 4.3. Surgery for CDDO-Me, SN50 or PDI siRNA Infusion

Control and epilepsy rats were infused (1) vehicle, (2) CDDO-Me (10 μM), (3) SN50 (an NF-κB inhibitor; 20 μM), (4) non-targeting control siRNA (GCAACUAACUUCGUUAGAAUCGUUAUU) or (5) PDI siRNA (UCUGCCUUCAGUUUUGCAGUU) into the right lateral ventricle (1 mm posterior; 1.5 mm lateral; −3.5 mm depth to the bregma) with a brain infusion kit 1 and an Alzet 1007D osmotic pump (Alzet, Cupertino, CA, USA) [22,72,73,74]. Each compound and siRNA could not evoke behavioral and neurological defects in animals.

### 4.4. Tissue Processing

Seven days after surgery (infusion), rats were perfused with 4% paraformaldehyde through ascending aorta under urethane anesthesia (1.5 g/kg i.p.). After cryoprotection with PB containing 30% sucrose, 30 μm thick sections were made using a cryostat. For Western blot, animals were decapitated under the same anesthesia (1.5 g/kg, i.p.). The brains were quickly dissected to 1 mm thickness using rodent brain matrix (World Precision Instruments, Sarasota, FL, United States). Thereafter, the stratum radiatum of the CA1 region of the dorsal hippocampus were rapidly obtained. The CA1 tissues were lysed and protein concentration determined using a Micro BCA Protein Assay Kit (Pierce Chemical, Rockford, IL, United States) [11,75].

### 4.5. Immunohistochemistry, Cell Counts, and Mitochondrial Morphometry

After blocking with 0.1% bovine serum albumin, sections were incubated with primary antibodies (Table 1) and Cy2- or Cy3-conjugated secondary antibodies. Immunofluorescence was observed using AxioScope microscope (Carl Zeiss Korea, Seoul, South Korea). A negative control test was performed with normal rabbit serum (#31883, ThermoFisher Korea, Seoul, South Korea), mouse IgG1 isotype control (#02-6100, ThermoFisher Korea, Seoul, South Korea), or mouse IgG2a isotype control (#02-6200, ThermoFisher Korea, Seoul, South Korea). Cell counts were performed using AxioVision Rel. 4.8 Software. The areas of interest (1 × 10^4^ μm^2^) in a section (10 sections per each animal) were selected from the CA1 region, and 30 areas/rat (400 μm^2^/area) in the stratum radiatum of CA1 region (15 sections from each animal, *n* = 7 in each group) were selected and fluorescent intensity was measured using AxioVision Rel. 4.8 software (Carl Zeiss Korea, Seoul, South Korea). In addition, 5 hippocampal sections from each animal were randomly selected and mitochondrial morphometry was performed in CA1 astrocytes from each slice (total 35 cells in each group, respectively) using ImageJ software. Thereafter, mitochondrial parameters (area-weighted form factor, form factor and cumulative area/perimeter ratio) were calculated [28,29]. The morphological analyses were conducted by investigators were blinded to experimental groups [11,76].

### 4.6. Western Blot

Sample proteins (10 μg) were loaded on a Bis-Tris sodium dodecyl sulfate-poly-acrylamide gel. After electrophoresis and transfer, membranes were incubated in primary antibodies (Table 1). The visualization and quantification of immunoband were conducted using ImageQuant LAS4000 system (GE Healthcare Korea, Seoul, South Korea). Density of immunobands was calibrated with the β-actin.

### 4.7. Measurement of SNO-PDI

The quantification of SNO-PDI performed using the S-nitrosylation Western Blot Kit (ThermoFisher). Lysates were labeled with iodoTMTzero reagents after MMT pretreatment. Thereafter, TMT-labeled proteins were precipitated by anti-TMT resin, eluted by TMT elusion buffer, and identified by Western blot. For controls, ascorbate was eliminated from each sample [22,74].

### 4.8. Data Analysis

The values on normality were determined by Shapiro–Wilk *W*-test. For comparisons of data, Student *t*-test or one-way ANOVA were performed. The post hoc test was performed with Bonferroni’s test. A *p*-value less than 0.05 was considered to be significant.

## 5. Conclusions

The present data reveal that the increased NF-κB S529 phosphorylation reduced the PDI protein level in clasmatodendritic CA1 astrocytes within the hippocampus of chronic epilepsy rats. Furthermore, clasmatodendritic CA1 astrocytes showed the accumulation of elongated mitochondria. CDDO-Me and SN50 attenuated clasmatodendrosis in CA1 astrocytes accompanied by the reduced NF-κB S529 phosphorylation and facilitated mitochondrial fission in CA1 astrocytes by increasing total PDI protein and SNO-PDI levels. PDI knockdown led to mitochondrial elongation in intact CA1 astrocytes under physiological conditions. Therefore, our findings suggest that NF-κB-mediated PDI downregulation may trigger clasmatodendrosis by abrogating DRP1-mediated mitochondrial fission, which is mitigated by CDDO-Me (Figure 9).

## Figures and Tables

**Figure 1 ijms-24-05875-f001:**
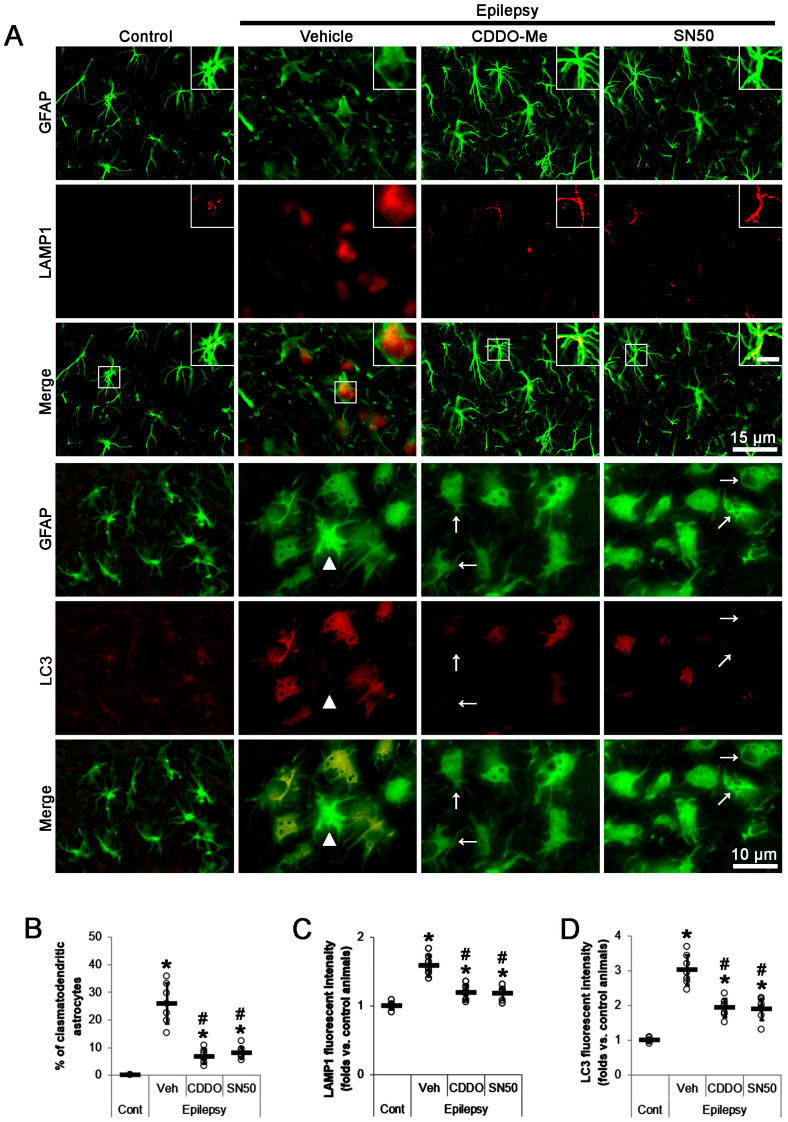
Effects of CDDO-Me and SN50 on clasmatodendritic changes in CA1 astrocytes of epilepsy rats. Compared to control animals (Cont), lysosomal-associated membrane protein 1 (LAMP1) and microtubule-associated protein 1A/1B light-chain 3 (LC3) expressions are upregulated in CA1 astrocytes in epilepsy rats. LC3 signal is undetectable in reactive astrocyte (arrowhead). Compared to vehicle (Veh), CDDO-Me (CDDO) and SN50 reduce the increased LAMP1 expression in CA1 astrocytes. In addition, they decrease the LC3 expression in some vacuolized CA1 astrocytes (arrows). (**A**) Representative photos demonstrating clasmatodendritic degeneration in CA1 astrocytes. Inserts are high magnification of rectangles (bar = 7.5 μm). (**B**–**D**) Quantifications of the protective effects of CDDO-Me and SN50 on clasmatodendrosis (**B**), LAMP1 upregulation (**C**), and LC3 induction in CA1 astrocytes (**D**). Open circles and horizontal bars indicate the individual value and the mean value, respectively. Error bars are S.D. (***, *^#^ p* < 0.05 vs. control rats and vehicle-treated epilepsy rats; *n* = 7).

**Figure 2 ijms-24-05875-f002:**
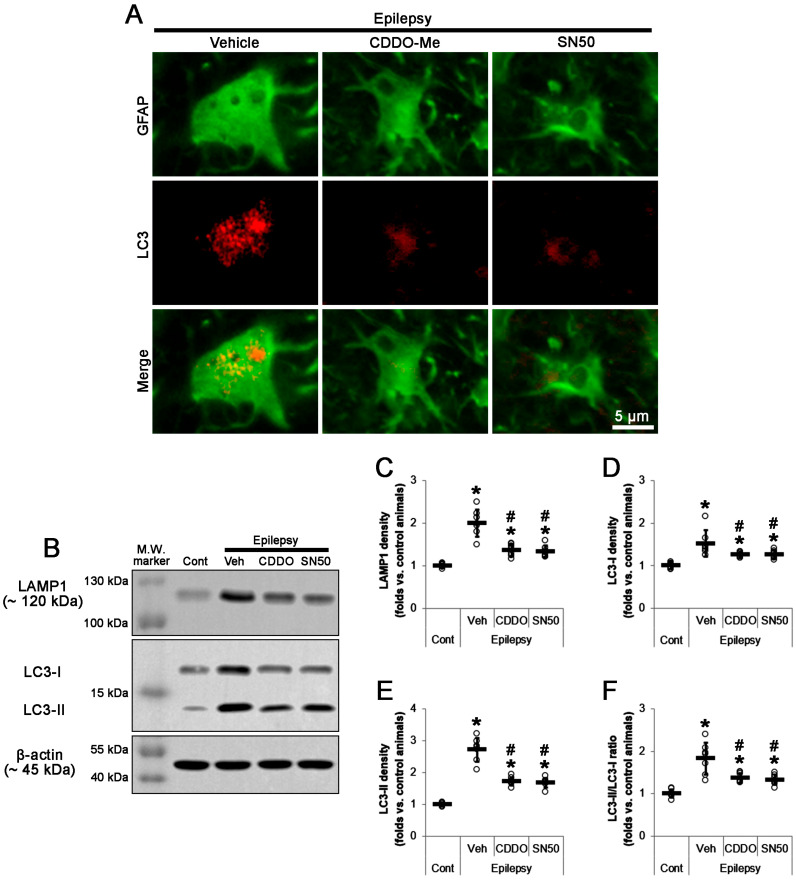
Effects of CDDO-Me and SN50 on autophagy process in the epileptic hippocampus. In vehicle-treated epilepsy rats, LC3-positive puncta are clearly observed in vacuolized astrocytes. Compared to control animals, LAMP1, LC3-I, and LC3-II densities are increased in the epileptic animals concomitant with the elevated LC3-II/LC3-I ratio, which are ameliorated by CDDO-Me and SN50. (**A**) Representative photos demonstrating LC3 immunoreactivity in clasmatodendritic CA1 astrocytes. (**B**) Representative Western blot images demonstrating LAMP1, LC3-I, and LC3-II bands in the epileptic hippocampus. (**C**,**D**) Quantifications of the effects of CDDO-Me and SN50 on LAMP1 (**C**), LC3-I (**D**), LC3-II (**E**), and LC3-II/LC3-I ratio (**F**). Open circles and horizontal bars indicate the individual value and the mean value, respectively. Error bars are S.D. (***, *^#^ p* < 0.05 vs. control rats and vehicle-treated epilepsy rats; *n* = 7).

**Figure 3 ijms-24-05875-f003:**
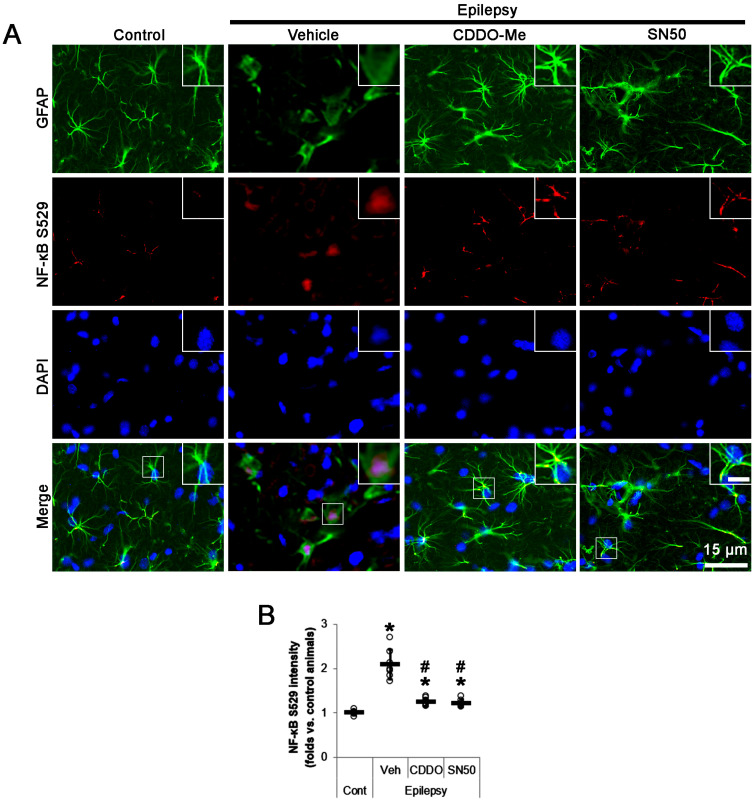
Effects of CDDO-Me and SN50 on NF-κB S529 phosphorylation in CA1 astrocytes. CDDO-Me (CDDO) and SN50 reduce the increased NF-κB S529 phosphorylation in CA1 astrocytes of chronic epilepsy rats compared to vehicle (Veh). (**A**) Representative photos demonstrating NF-κB S529 phosphorylation in CA1 astrocytes. Inserts are high magnification of rectangles (bar = 7.5 μm). (**B**) Quantifications of the effects of CDDO-Me and SN50 on NF-κB S529 level in CA1 astrocytes. Open circles and horizontal bars indicate the individual value and the mean value, respectively. Error bars are S.D. (***, *^#^ p* < 0.05 vs. control and vehicle-treated epilepsy rats, respectively; *n* = 7).

**Figure 4 ijms-24-05875-f004:**
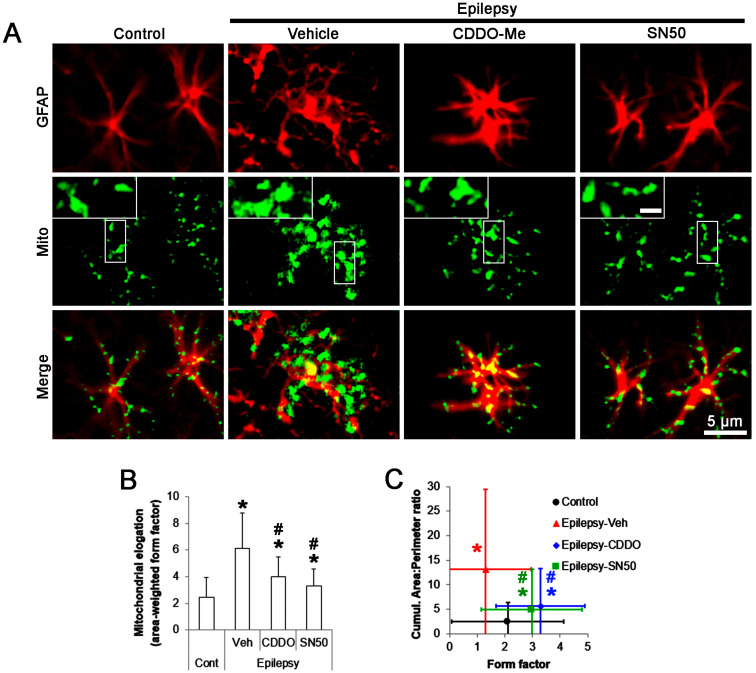
Effects of CDDO-Me and SN50 on mitochondrial length in CA1 astrocytes. CDDO-Me (CDDO) and SN50 ameliorate mitochondrial hyperfusion in CA1 astrocytes compared to vehicle (Veh). (**A**) Representative photos demonstrating mitochondrial morphology in CA1 astrocytes. (**B**,**C**) Quantifications of mitochondrial elongation index (mean ± S.D.; ***, *^#^ p* < 0.05 vs. control and vehicle-treated epilepsy rats, respectively; *n* = 7).

**Figure 5 ijms-24-05875-f005:**
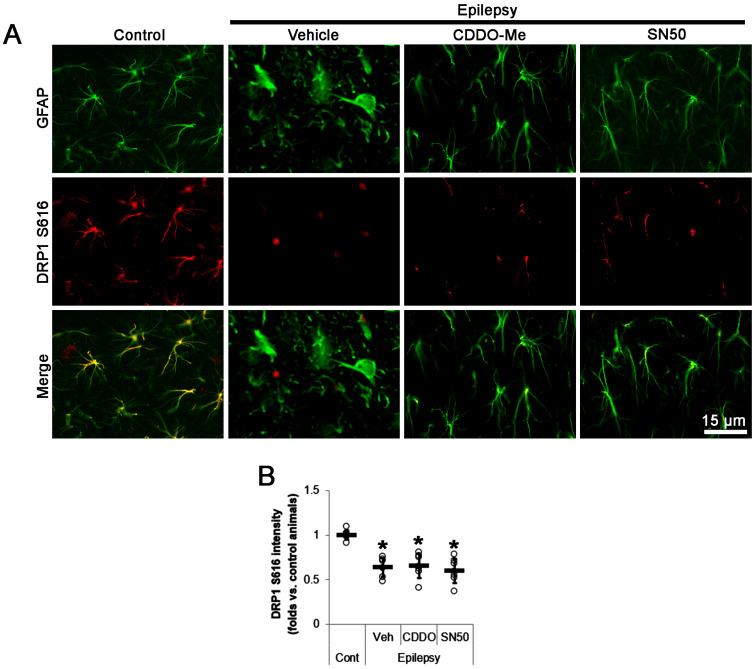
Effects of CDDO-Me and SN50 on dynamin-related protein 1 (DRP1) S616 phosphorylation in CA1 astrocytes of epilepsy rats. DRP1 S616 expression is reduced in CA1 astrocytes in epilepsy rats, which is unaffected by CDDO-Me and SN50. (**A**) Representative photos demonstrating DRP1 S616 phosphorylation in CA1 astrocytes. (**B**) Quantifications of DRP1 S616 level in CA1 astrocytes. Open circles and horizontal bars indicate the individual value and the mean value, respectively. Error bars are S.D. (** p* < 0.05 vs. control and vehicle-treated epilepsy rats; *n* = 7).

**Figure 6 ijms-24-05875-f006:**
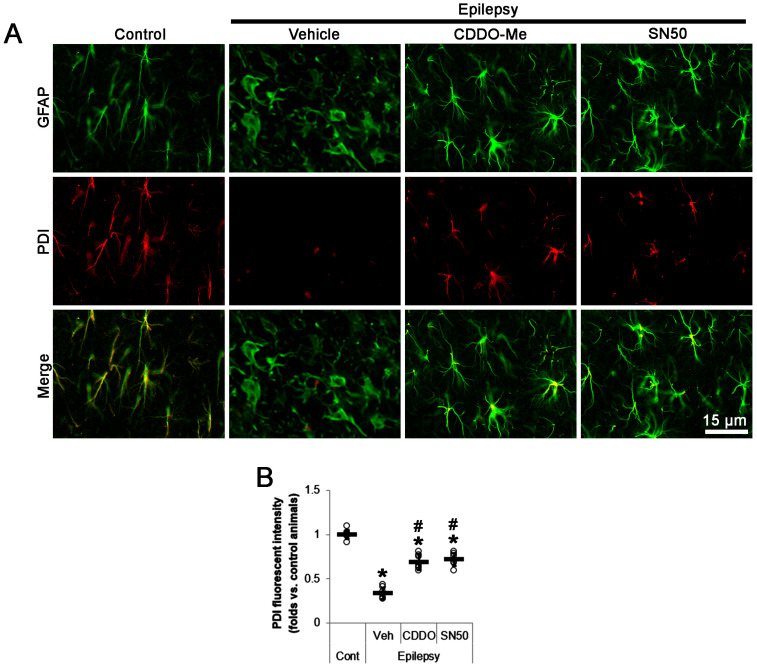
Effects of CDDO-Me and SN50 on protein disulfide isomerase (PDI) expression in CA1 astrocytes of epilepsy rats. PDI expression is decreased in CA1 astrocytes, which is enhanced by CDDO-Me and SN50. (**A**) Representative photos demonstrating PDI expression in CA1 astrocytes. (**B**) Quantifications of the effects of CDDO-Me and SN50 on PDI expression in CA1 astrocytes. Open circles and horizontal bars indicate the individual value and the mean value, respectively. Error bars indicate S.D. (***, *^#^ p* < 0.05 vs. control and vehicle-treated epilepsy rats; *n* = 7).

**Figure 7 ijms-24-05875-f007:**
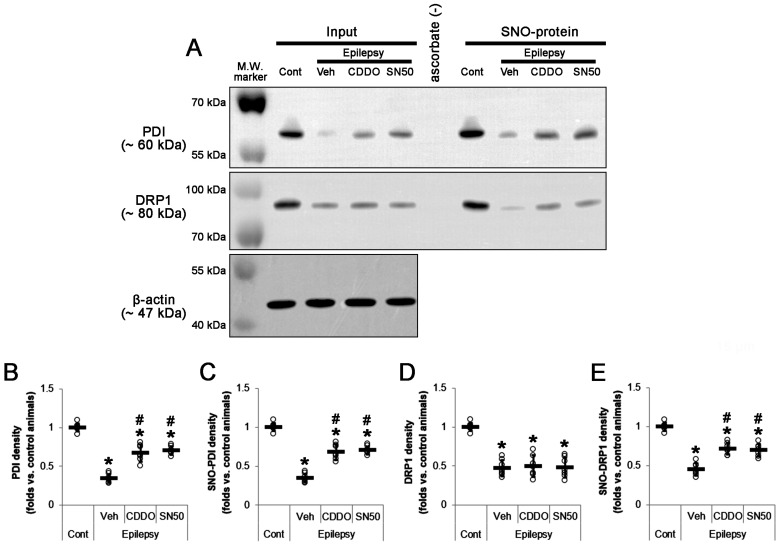
Effects of CDDO-Me and SN50 on total and S-nitrosylated (SNO-) levels of PDI and DRP1 in the stratum radiatum of the CA1 region of epilepsy rats. Total and S-nitrosylated (SNO-) levels of PDI and DRP1 are decreased in CA1 astrocytes in epilepsy rats, which are enhanced by CDDO-Me and SN50. (**A**) Representative Western blot images demonstrating total and SNO levels of PDI and DRP1. (**B**–**E**) Quantifications of the effects of CDDO-Me and SN50 on total PDI (**B**), SNO-PDI (**C**), total DRP1 (**D**), and SNO-DRP1 (**E**) levels. Open circles and horizontal bars indicate the individual value and the mean value, respectively. Error bars are S.D. (***, *^#^ p* < 0.05 vs. vehicle; *n* = 7).

**Figure 8 ijms-24-05875-f008:**
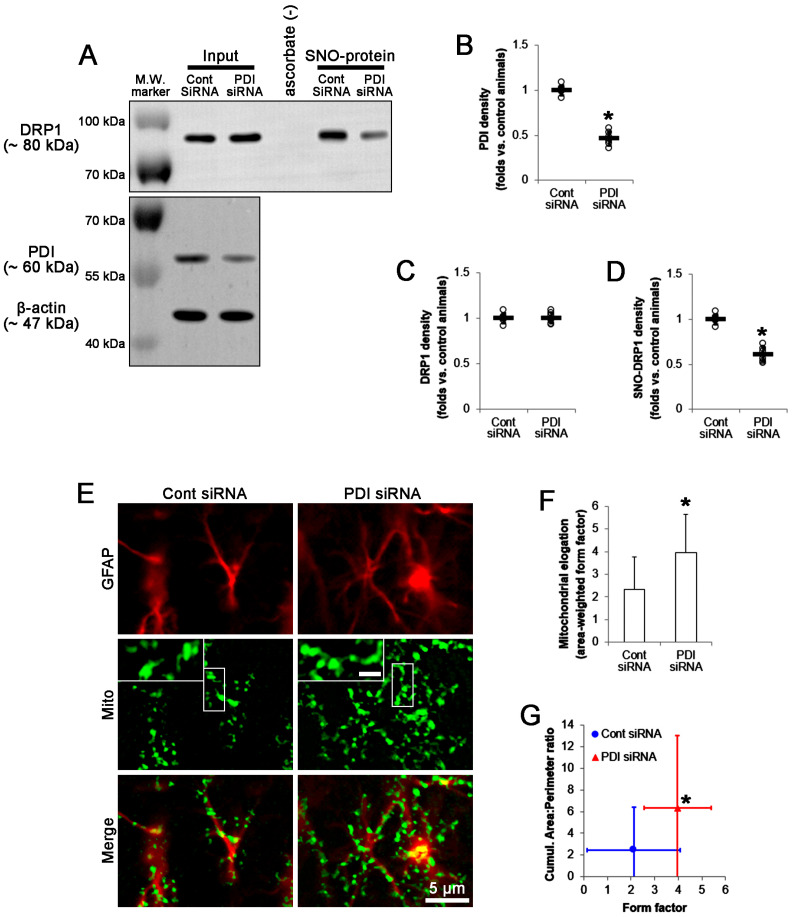
Effects of PDI knockdown on total and S-nitrosylated (SNO-) DRP1 levels and mitochondrial elongation in the stratum radiatum of the CA1 region of normal rats. Compared to control siRNA (Cont siRNA), PDI siRNA reduces total PDI and SNO-DRP1 levels without changing total DRP1 level and elongates mitochondrial length in CA1 astrocytes. (**A**) Representative Western blot images of demonstrating total and SNO-DRP1 levels. (**B**–**D**) Quantifications of total PDI (**B**), total DRP1 (**C**), and SNO-DRP1 (**D**) levels. Open circles and horizontal bars indicate the individual value and the mean value, respectively. Error bars are S.D. (** p* < 0.05 vs. control siRNA; *n* = 7). (**E**) Representative photos demonstrating mitochondrial morphology in CA1 astrocytes. (**F**,**G**) Quantifications of the effects of PDI knockdown on mitochondrial elongation index (mean ± S.D.; ** p* < 0.05 vs. control siRNA; *n* = 7).

**Figure 9 ijms-24-05875-f009:**
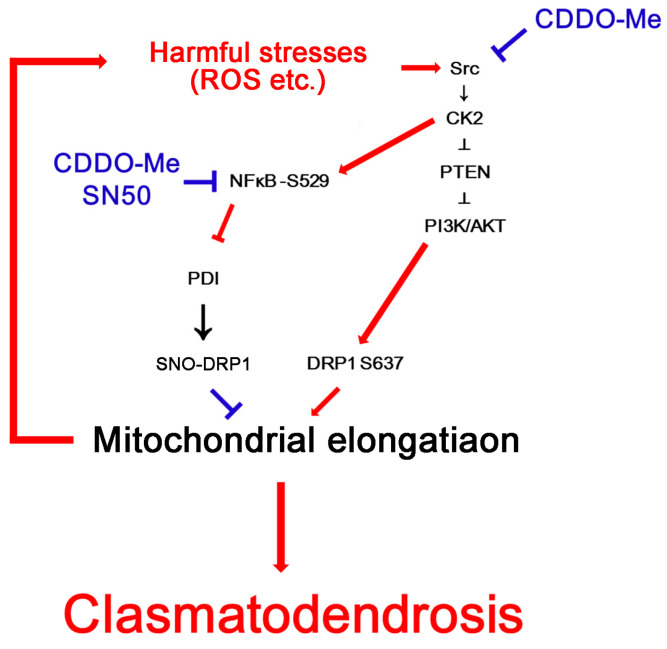
Schematic depiction representing the effects of CDDO-Me and SN50 on clasmaotodendrosis based on the present data and previous reports [11,20]. CDDO-Me inhibits Src-casein kinase 2 (CK2)-tensin homolog deleted on chromosome 10 (PTEN)-phosphatidylinositol-3-kinase (PI3K)/AKT-mediated DRP1 S637 phosphorylation, which elongates mitochondrial length. In addition, both CDDO-Me and SN50 abrogate CK2-mediated NF-κB S529 phosphorylation and facilitate mitochondrial fission by enhancing PDI-mediated DRP1 S-nitrosylation. These modes of regulation prevent clasmatodendritic degeneration of CA1 astrocytes.

**Table 1 ijms-24-05875-t001:** Primary antibodies used in the present study.

Antigen	Host	Manufacturer (Catalog Number)	Dilution Used
DRP1	Rabbit	Thermo (#PA1-16987)	1:1000 (WB)
DRP1 S616	Rabbit	Cell Signaling (#4494)	1:200 (IH)
GFAP	Rabbit Mouse	Abcam (#ab7260) Millipore (#MAB3402)	1:500 (IH) 1:2000 (IH)
LAMP1	Rabbit	Lifespan (#LS-B580)	1:200 (IH) 1:1000 (WB)
LC3	Rabbit	Abgent (#AP1802a)	1:100 (IH) 1:1000 (WB)
Mitochondrial marker (Mitochondrial complex IV subunit 1, MTCO1)	Mouse	Abcam (#ab14705)	1:500 (IH)
NF-κB S529	Rabbit	Abcam (#ab47395)	1:100 (IH)
PDI	Mouse	Abcam (#ab2792)	1:100 (IH) 1:1000 (WB)
β-actin	Mouse	Sigma (#A5316)	1:6000 (WB)

IH: immunohistochemistry; WB, Western blot.

## Data Availability

Not applicable.

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
