# Peer review of "CDDO-Me Abrogates Aberrant Mitochondrial Elongation in Clasmatodendritic Degeneration by Regulating NF-κB-PDI-Mediated S-Nitrosylation of DRP1"

_ijms, 2023, doi:10.3390/ijms24065875_

Round 1

Reviewer 1 Report

This work is aimed at evaluating the therapeutic efficacy of CDDO-Me and SN-50 on clasmatodendrosis in the hippocampus of chronic epilepsy rat model. The main results demonstrated that clasmatodendrosis in indeed ameliorated by the administration of the two compounds. This event is accompanied by improved mitochondrial fission.

The work is characterized by several shortcomings.

1) The same research group already published in IJMS a research paper focused on the putative role of CDDO-Me in chronic epilepsy-induced clasmatodendrosis. Even though the molecular targets/pathways are different, the overlap between the two  works undermines the novelty of the this manuscript, which is only partial.

2) The authors often refer to clasmatodendrosis as an autophagic death. In my opinion this definition is not coherent, as several works demonstrated that autophagy induction is only one of the features characterizing clasmatodendrosis. Notably, clasmatodendrosis is also associated to UPS alterations, inflammatory status and osmotic stress. According to this notion, autophagy inhibitors only partially mitigate, but not entirely reverse, clasmatodendrosis in both in vitro and in vivo models.

3) As already stated, the authors consider clasmatodendrosis as an autophagic cell death. However, autophagy is not assessed in this work. This strongly limits the strength of the work, since it should be important to evaluate how the compounds used in this work may restore autophagy alterations.

4) The authors assessed the potential pharmacological efficacy of CDDO-Me, a Nrf-2 activator, and SN-50, a NFkB inhibitor. To improve the validiy of the experimental plan, a combined therapy should be shown.

5) Data about LAMP1 are not robust. LAMP1 staining is peculiar because of its lysosomal localization. In this work LAMP1 immunoreactivity does not reflect organelle staining. Furthermore, LAMP1 is ubiquitously expressed at basal levels, even in astrocytes. Thus, it is unusual that LAMP1 staining is completely absent in the hippocampus of control animals.

6) Data are expressed as the mean ± S.E.M. The standard error of the mean indicates the uncertainty of how the sample mean represents the population mean. In my opinion, the authors inappropriately report the SEM instead of the Standard Deviation (SD). Since the SEM is always less than the SD, it deceives the reader into underestimating the variability between individuals within the study sample.

Author Response

Dear, Reviewer 1;

I am enclosing herewith the revised version of our manuscript entitled CDDO-Me abrogates aberrant mitochondrial elongation in au-tophagic astroglial death (clasmatodendrosis) by regulating NF-κB-PDI-mediated S-nitrosylation of DRP1 (Ms. No.: ijms-2217732)”.

We appreciate the reviewer’s recommendations to improve our manuscript.

With respect to reviewers’ comments and suggestions, we addressed the reviewers' points.

My responses to comments are as followed;

Reviewer 1

1) The same research group already published in IJMS a research paper focused on the putative role of CDDO-Me in chronic epilepsy-induced clasmatodendrosis. Even though the molecular targets/pathways are different, the overlap between the two  works undermines the novelty of the this manuscript, which is only partial.

Answer: Clasmatodendrosis is one of the irreversible astroglial degeneration, which is involved in seizure duration and its progression in the epileptic hippocampus. Although sustained heat shock protein 25 (HSP25) induction leads to this autophagic astroglial death, dysregulation of mitochondrial dynamics (aberrant mitochondrial elongation) is also involved in the pathogenesis in clasmatodendrosis. In a previous study, therefore, we had investigated the relevance between HSP25 upregulation and mitochondrial hyperfusion in clasmatodendrosis, and suggested that the dysregulation of HSP25-AKT-DRP1-mediated mitochondrial dynamics may play an important role in clasmatodendrosis, which had been abrogated by CDDO-Me.

On the other hand, the enhanced NFκB S529 phosphorylation also leads to clasmatodendrosis via unknown mechanism. Since CDDO-Me inhibits NFκB, it is likely that NFκB-mediated signaling pathway may be involved in clasmatodendritic degeneration induced by impaired mitochondrial dynamics, which has been unexplored. In the present study, therefore, we investigated the role of NFκB S529 phosphorylation and its downstream effector in impaired mitochondrial dynamics in clasmatodendrosis, and found that NFκB S529 phosphorylation led to mitochondrial hyperfusion in clasmatodendritic astrocytes by inhibiting PDI-mediated S-nitrosylation of DRP1. Therefore, the present manuscript is totally novel in the goal of study, the applied techniques, the targeting signaling pathway, chemicals/siRNA used in the study and underlying mechanism of impaired mitochondrial dynamics, as compared to the previous one. We summarize the novelty as the Table below;

The present study

The previous studies

The Goal of the study

To elucidate the role of NFκB S529 phosphorylation and its downstream effector in impaired mitochondrial dynamics in clasmatodendrosis.

To elucidate the role of HSP25-AKT-DRP1 signaling axis in impaired mitochondrial dynamics in clasmatodendrosis.

Targeting signaling pathway

NFκB, PDI, DRP1, NO (S-nitrosylation)

HSP25, AKT, DRP1

Chemicals/siRNA used in the study

CDDO-Me, SN50, PDI siRNA

CDDO-Me, 3CAI, HSP25 siRNA

Techniques

Immunohistochemistry, Western blot, S-nitrosylation assay

Immunohistochemistry, Western blot

Underlying mechanism and Scientific significance

NFκB S529 phosphorylation led to mitochondrial hyperfusion in clasmatodendritic astrocytes by inhibiting PDI-mediated S-nitrosylation of DRP1.

HSP25-AKT signaling pathway resulted in hyperfusion/aggregation of mitochondria by DRP1 S637 phosphorylation

To the best of our knowledge, therefore, our findings suggest, for the first time, that NFκB-mediated PDI downregulation may trigger autophagic astroglial degeneration by abrogating DRP1-mediated mitochondrial fission.

2) The authors often refer to clasmatodendrosis as an autophagic death. In my opinion this definition is not coherent, as several works demonstrated that autophagy induction is only one of the features characterizing clasmatodendrosis. Notably, clasmatodendrosis is also associated to UPS alterations, inflammatory status and osmotic stress. According to this notion, autophagy inhibitors only partially mitigate, but not entirely reverse, clasmatodendrosis in both in vitro and in vivo models.

Answer: As reviewer’s comments, clasmatodendrosis is relevant to UPS alterations, inflammatory status and osmotic stress. However, UPS alteration, inflammation and osmotic stress lead to autophagy. Thus, it is likely that clasmatodendrosis is one of the autophagic astroglial degeneration of which the underlying mechanisms are very complicated. This is also the reason why we continued to investigate clasmatodendrosis.

3) As already stated, the authors consider clasmatodendrosis as an autophagic cell death. However, autophagy is not assessed in this work. This strongly limits the strength of the work, since it should be important to evaluate how the compounds used in this work may restore autophagy alterations.

Answer: Basically, autophagy is vital process for cell survivals. However, excessive autophagy evokes type II programmed cell death, known as autophagic cell death (Bursch et al., J Cell Sci. 2000 Apr;113 ( Pt 7):1189-98. doi: 10.1242/jcs.113.7.1189). Indeed, we have reported that clasmatodendrosis is correlated the reduced number of astroglial population in the stratum radiatum of CA1 region (Kim et al., J Comp Neurol. 2008 Dec 10;511(5):581-98. doi: 10.1002/cne.21851; Kim et al., Hippocampus. 2011 Dec;21(12):1318-33. doi: 10.1002/hipo.20850; Kim et al., Front Cell Neurosci. 2017 Feb 22;11:47. doi: 10.3389/fncel.2017.00047). According to other reviewer’s comments, furthermore, we have added the additional data concerning the effect of CDDO-Me and SN50 on expressions of microtubule-associated protein 1A/1B-light chain 3 (LC3, a credible autophagy maker that is bound to the membrane of autophagosome). Compatible with our previous study (Ryu et al., p65/RelA-Ser529 NF-κB subunit phosphorylation induces autophagic astroglial death (Clasmatodendrosis) following status epilepticus. Cell Mol Neurobiol. 2011 Oct;31(7):1071-8. doi: 10.1007/s10571-011-9706-1), LC3 expression was upregulated in most of clasmatodendritic astrocytes, which were attenuated by SN50 and CDDO-Me. Thus, both CDDO-Me and SN50 decreased the LC3 expression in some vacuolized CA1 astrocytes. These findings can support that the effect of CDDO-Me and SN50 on the autophagic mechanisms in clasmatodendritic CA1 astrocytes. We have inserted these data in Fig. 1A and D.

4) The authors assessed the potential pharmacological efficacy of CDDO-Me, a Nrf-2 activator, and SN-50, a NFkB inhibitor. To improve the validiy of the experimental plan, a combined therapy should be shown.

Answer: As described in the text, CDDO-Me directly inhibits NF-κB signaling pathway, while it activates Nrf-2. Thus, we applied SN50 to directly inhibit NF-κB-mediated signaling pathway without affecting Nrf-2 activity. We will appreciate if reviewer would consider these.

5) Data about LAMP1 are not robust. LAMP1 staining is peculiar because of its lysosomal localization. In this work LAMP1 immunoreactivity does not reflect organelle staining. Furthermore, LAMP1 is ubiquitously expressed at basal levels, even in astrocytes. Thus, it is unusual that LAMP1 staining is completely absent in the hippocampus of control animals.

Answer: LAMP1 expression level is very low and undetectable in astrocytes in vivo. Furthermore, clasmatodendritic degeneration is also characterized by vacuolization in cell bodies. Indeed, vacuoles in clasmatodendritic astrocytes were divided into three groups (Ryu et al., Brain Res Bull. 2011 Jul 15;85(6):368-73. doi: 10.1016/j.brainresbull.2011.05.007): One had large-sized vacuoles (diameter > 2 μm), round-shaped cell body, short blunt processes and GFAP tangles in the cytoplasm. Based on these morphological evidences, astrocytes containing large vacuoles were clasmatodendritic astrocytes. Another group had medium-sized vacuoles (1 μm < diameter < 2 μm) that were evenly widespread through cytoplasm. Medium-sized vacuoles were in the peripheral regions and processes. Some vacuoles were attached to the cell membrane. Based on these morphological evidences, astrocytes containing medium vacuoles were degenerating astrocytes. The third group had small-sized vacuoles (diameter < 0.5 μm) that were clearly observed in long-curled processes. Some vacuoles were attached to other vacuoles or the membrane. Therefore, some vacuoles seemed like docking and fusions with each other. In addition, vacuoles showed LAMP1 immunoreactivity, but not GM130 (a maker for Golgi apparatus) immunoreactivity. Furthermore, vacuoles contained LC3-II (a marker for autophagy) immunoreactivity. Thus, the increase in the size and the number of LAMP1 vacuoles indicates that clasmatodendrosis is lysosome-derived autophagic astroglial degeneration. Therefore, we believe that LAMP1 upregulation is a useful marker for identification of clasmatodendrosis, and have applied a lot of studies.

6) Data are expressed as the mean ± S.E.M. The standard error of the mean indicates the uncertainty of how the sample mean represents the population mean. In my opinion, the authors inappropriately report the SEM instead of the Standard Deviation (SD). Since the SEM is always less than the SD, it deceives the reader into underestimating the variability between individuals within the study sample.

Answer: SEM can directly represent the statistical significance among groups, which SD cannot. Thus, we believe that SEM is suitable to express data rather than SD. Furthermore, we have also reported an individual value in graphs, which would help the reader to underestimate the between individuals within the study sample.

I will be grateful if the manuscript could be reviewed and considered for publication.

Best Regards,

Ji-Eun Kim, Ph.D

Department of Anatomy and Neurobiology

Hallym University, College of Medicine

Chunchon, 24252, Kangwon-Do

Republic of Korea

Reviewer 2 Report

The manuscript by Duk Shin Le explored the mechanisms of impaired mitochondrial dynamics during clasmatodendrosis. In particular, they suggest that NF-.B-mediated PDI downregulation may trigger autophagic astroglial degeneration by abrogating DRP1-mediated mitochondrial fission, which is mitigated by CDDO-Me".

The study is interesting and well-organized. The results showing mitochondrial elongation are clear and convincing.  Other aspects need to be improved.

Major points

The authors have to show the effect of CDDO-Me and also SN50 molecules on the autophagic mechanisms in clasmatodendritic CA1 astrocytes.

The authors have to improve the quality of the immunofluorescence images.  I suggest including the high magnification at least in Figure 2 A. 

Author Response

Dear, Reviewer 2;

I am enclosing herewith the revised version of our manuscript entitled CDDO-Me abrogates aberrant mitochondrial elongation in au-tophagic astroglial death (clasmatodendrosis) by regulating NF-κB-PDI-mediated S-nitrosylation of DRP1 (Ms. No.: ijms-2217732)”.

We appreciate the reviewer’s recommendations to improve our manuscript.

With respect to reviewers’ comments and suggestions, we addressed the reviewers' points.

My responses to comments are as followed;

Reviewer 2

The manuscript by Duk Shin Le explored the mechanisms of impaired mitochondrial dynamics during clasmatodendrosis. In particular, they suggest that NF-.B-mediated PDI downregulation may trigger autophagic astroglial degeneration by abrogating DRP1-mediated mitochondrial fission, which is mitigated by CDDO-Me. The study is interesting and well-organized. The results showing mitochondrial elongation are clear and convincing. Other aspects need to be improved.

Major points

  1. The authors have to show the effect of CDDO-Me and also SN50 molecules on the autophagic mechanisms in clasmatodendritic CA1 astrocytes.

Answer: Does this comment mean that the capacities of CDDO-Me and SN50 on autophagic markers should be validated?

IF YES, we have added the additional data concerning the effect of CDDO-Me and SN50 on expressions of microtubule-associated protein 1A/1B-light chain 3 (LC3, a credible autophagy maker that is bound to the membrane of autophagosome). Compatible with our previous study (Ryu et al., p65/RelA-Ser529 NF-κB subunit phosphorylation induces autophagic astroglial death (Clasmatodendrosis) following status epilepticus. Cell Mol Neurobiol. 2011 Oct;31(7):1071-8. doi: 10.1007/s10571-011-9706-1), LC3 expression was upregulated in most of clasmatodendritic astrocytes, which were attenuated by SN50 and CDDO-Me. Thus, both CDDO-Me and SN50 decreased the LC3 expression in some vacuolized CA1 astrocytes. These findings can support that the effect of CDDO-Me and SN50 on the autophagic mechanisms in clasmatodendritic CA1 astrocytes. We have inserted these data in Fig. 1A and D.

IF NO, the present study demonstrates that both CDDO-Me and SN50 effectively attenuated clasmatodendrosis of CA1 astrocytes by inhibiting NFκB-PDI-DRP1-medaited mitochondrial elongation (Fig. 8). Furthermore, both CDDO-Me and SN50 mitigated LAMP-1 expression in clasmatodendritic CA1 astrocytes. Thus, we believe that these data can represent the effect of CDDO-Me and SN50 on the autophagic mechanisms in clasmatodendrosis in CA1 astrocytes.

  1. The authors have to improve the quality of the immunofluorescence images. I suggest including the high magnification at least in Figure 2 A. 

Answer: With respect to reviewer’s comments, we have inserted the high magnification photos in Figure 2A.

I will be grateful if the manuscript could be reviewed and considered for publication.

Best Regards,

Ji-Eun Kim, Ph.D

Department of Anatomy and Neurobiology

Hallym University, College of Medicine

Chunchon, 24252, Kangwon-Do

Republic of Korea

Reviewer 3 Report

International Journal of Molecular Sciences (Manuscript ID: ijms-2217732), Comments to the Authors:

Title: CDDO-Me abrogates aberrant mitochondrial elongation in au-tophagic astroglial death (clasmatodendrosis) by regulating NF-κB-PDI-mediated S-nitrosylation of DRP1

Comments

The submitted manuscript discussed the effect of 2-cyano-3,12-dioxo-oleana-1,9(11)-dien-28-oic acid methyl ester (CDDO-Me; bardoxolone methyl or RTA 402) and SN50 (a nuclear factor-κB (NF-κB) inhibitor) on clasmatodendrosis in CA1 astrocytes in the hippocampus of chronic epilepsy rats accompanied by the reduced NF-κB S529 phosphorylation. CDDO-Me and SN50 facilitated mitochondrial fission in CA1 astrocytes, independent of dynamin-related protein 1 (DRP1) S616 phosphorylation. Both CDDO-Me and SN50 increased total PDI protein and S-nitrosylated (SNO)-PDI levels in CA1 astrocytes of chronic epilepsy rats. PDI knockdown resulted in mitochondrial elongation in intact CA1 astrocytes under physiological condition, while it did not evoke clasmatodendrosis.  

I think the submitted manuscript can be accepted for publication after the authors respond to the following comments:

  1. The abstract should be rephrased because it is disorganized and have several introductory sentences. The abstract should contain numerical numbers so the readers can evaluate and compare the results of previous publications and this work.
  2. The authors published 5 papers on this topic. They have to compare and clrealy indicate the differences between the current work and their previous publications including A) Kim, Ji-Eun, Hana Park, and Tae-Cheon Kang. 2023. "Peroxiredoxin 6 Regulates Glutathione Peroxidase 1-Medited Glutamine Synthase Preservation in the Hippocampus of Chronic Epilepsy Rats" Antioxidants 12, no. 1: 156. https://doi.org/10.3390/antiox12010156; B) Kim, Ji-Eun, Duk-Shin Lee, and Tae-Cheon Kang. 2022. "Sp1-Mediated Prdx6 Upregulation Leads to Clasmatodendrosis by Increasing Its aiPLA2 Activity in the CA1 Astrocytes in Chronic Epilepsy Rats" Antioxidants 11, no. 10: 1883. https://doi.org/10.3390/antiox11101883; C) Lee, Duk-Shin, Tae-Hyun Kim, Hana Park, and Ji-Eun Kim. 2022. "CDDO-Me Attenuates Clasmatodendrosis in CA1 Astrocyte by Inhibiting HSP25-AKT Mediated DRP1-S637 Phosphorylation in Chronic Epilepsy Rats" International Journal of Molecular Sciences 23, no. 9: 4569. https://doi.org/10.3390/ijms23094569; D) Kim, Ji-Eun, Duk-Shin Lee, Tae-Hyun Kim, and Tae-Cheon Kang. 2022. "Glutathione Regulates GPx1 Expression during CA1 Neuronal Death and Clasmatodendrosis in the Rat Hippocampus following Status Epilepticus" Antioxidants 11, no. 4: 756. https://doi.org/10.3390/antiox11040756; E) Kim, Ji-Eun, and Tae-Cheon Kang. 2021. "CDDO-Me Attenuates Astroglial Autophagy via Nrf2-, ERK1/2-SP1- and Src-CK2-PTEN-PI3K/AKT-Mediated Signaling Pathways in the Hippocampus of Chronic Epilepsy Rats" Antioxidants 10, no. 5: 655. https://doi.org/10.3390/antiox10050655.
  3. The authors should rephrase the paper to indicate the effect of CDDO-Me and SN50. They should describe the effect of each agent separately. 

Author Response

Dear, Reviewer 3;

I am enclosing herewith the revised version of our manuscript entitled CDDO-Me abrogates aberrant mitochondrial elongation in au-tophagic astroglial death (clasmatodendrosis) by regulating NF-κB-PDI-mediated S-nitrosylation of DRP1 (Ms. No.: ijms-2217732)”.

We appreciate the reviewer’s recommendations to improve our manuscript.

With respect to reviewers’ comments and suggestions, we addressed the reviewers' points.

My responses to comments are as followed;

Reviewer 3

The submitted manuscript discussed the effect of 2-cyano-3,12-dioxo-oleana-1,9(11)-dien-28-oic acid methyl ester (CDDO-Me; bardoxolone methyl or RTA 402) and SN50 (a nuclear factor-κB (NF-κB) inhibitor) on clasmatodendrosis in CA1 astrocytes in the hippocampus of chronic epilepsy rats accompanied by the reduced NF-κB S529 phosphorylation. CDDO-Me and SN50 facilitated mitochondrial fission in CA1 astrocytes, independent of dynamin-related protein 1 (DRP1) S616 phosphorylation. Both CDDO-Me and SN50 increased total PDI protein and S-nitrosylated (SNO)-PDI levels in CA1 astrocytes of chronic epilepsy rats. PDI knockdown resulted in mitochondrial elongation in intact CA1 astrocytes under physiological condition, while it did not evoke clasmatodendrosis. I think the submitted manuscript can be accepted for publication after the authors respond to the following comments:

  1. The abstract should be rephrased because it is disorganized and have several introductory sentences. The abstract should contain numerical numbers so the readers can evaluate and compare the results of previous publications and this work.

Answer: With respect to reviewer’s comments, we have re-edited and re-written the abstract as followed;

“Clasmatodendrosis is a type II programmed cell death (autophagic cell death) of astrocytes. Although abnormal mitochondrial elongation is relevant to this astroglial degeneration, the underlying mechanisms of aberrant mitochondrial dynamics are still incompletely understood. Protein disulfide isomerase (PDI) is an oxidoreductase in the endoplasmic reticulum (ER). Since PDI expression is downregulated in clasmatodendritic astrocytes, PDI may be involved in aberrant mitochondrial elongation in clasmatodendritic astrocytes. In the present study, twenty-six percent of CA1 astrocytes showed clasmatodendritic degeneration in chronic epilepsy rats. 2-cyano-3,12-dioxo-oleana-1,9(11)-dien-28-oic acid methyl ester (CDDO-Me; bardoxolone methyl or RTA 402) and SN50 (a nuclear factor-κB (NF-κB) inhibitor) ameliorated the fraction of clasmatodendritic astrocytes to 6.8 and 8.1% in CA1 astrocytes, accompanied by the reduced lysosomal-associated membrane protein 1 (LAMP1) and microtubule-associated protein 1A/1B-light chain 3 (LC3) expressions. Furthermore, CDDO-Me and SN50 reduced the NF-κB S529 fluorescent intensity to 0.6 and 0.57-fold of vehicle-treated animal level, respectively. CDDO-Me and SN50 facilitated mitochondrial fission in CA1 astrocytes, independent of dynamin-related protein 1 (DRP1) S616 phosphorylation. In chronic epilepsy rats, total PDI protein, S-nitrosylated PDI (SNO-PDI) and SNO-DRP1 levels were 0.35-, 0.34- and 0.45-fold of control level in the CA1 region, which were increased CDDO-Me and SN50. Furthermore, PDI knockdown resulted in mitochondrial elongation in intact CA1 astrocytes under physiological condition, while it did not evoke clasmatodendrosis. Therefore, our findings suggest that NF-κB-mediated PDI inhibition may play an important role in autophagic astroglial degeneration via aberrant mitochondrial elongation.”

  1. The authors published 5 papers on this topic. They have to compare and clrealy indicate the differences between the current work and their previous publications including A) Kim, Ji-Eun, Hana Park, and Tae-Cheon Kang. 2023. "Peroxiredoxin 6 Regulates Glutathione Peroxidase 1-Medited Glutamine Synthase Preservation in the Hippocampus of Chronic Epilepsy Rats" Antioxidants 12, no. 1: 156. https://doi.org/10.3390/antiox12010156; B) Kim, Ji-Eun, Duk-Shin Lee, and Tae-Cheon Kang. 2022. "Sp1-Mediated Prdx6 Upregulation Leads to Clasmatodendrosis by Increasing Its aiPLA2 Activity in the CA1 Astrocytes in Chronic Epilepsy Rats" Antioxidants 11, no. 10: 1883. https://doi.org/10.3390/antiox11101883; C) Lee, Duk-Shin, Tae-Hyun Kim, Hana Park, and Ji-Eun Kim. 2022. "CDDO-Me Attenuates Clasmatodendrosis in CA1 Astrocyte by Inhibiting HSP25-AKT Mediated DRP1-S637 Phosphorylation in Chronic Epilepsy Rats" International Journal of Molecular Sciences 23, no. 9: 4569. https://doi.org/10.3390/ijms23094569 [20]; D) Kim, Ji-Eun, Duk-Shin Lee, Tae-Hyun Kim, and Tae-Cheon Kang. 2022. "Glutathione Regulates GPx1 Expression during CA1 Neuronal Death and Clasmatodendrosis in the Rat Hippocampus following Status Epilepticus" Antioxidants 11, no. 4: 756. https://doi.org/10.3390/antiox11040756; E) Kim, Ji-Eun, and Tae-Cheon Kang. 2021. "CDDO-Me Attenuates Astroglial Autophagy via Nrf2-, ERK1/2-SP1- and Src-CK2-PTEN-PI3K/AKT-Mediated Signaling Pathways in the Hippocampus of Chronic Epilepsy Rats" Antioxidants 10, no. 5: 655. https://doi.org/10.3390/antiox10050655.

Answer: With respect to reviewer’s comments, we have also discussed the differences between the current work and our previous publications as followed;

“Recently, we have reported that various enzymes regulate clasmatodendrosis. In particular, AKT, glutathione peroxidase 1 (GPx1) and peroxiredoxin 6 (Prdx6) play important roles in clasmatodendritic degeneration [11,20,66,67,68]. The dysregulation of Prdx6-GPx1-mediated signaling pathway exerts clasmatodendrosis by augmenting oxidative stress [66,67,68]. Oxidative stress also induces AKT activation that phosphorylates DRP1 S637 and facilitates bax-interacting factor 1 (Bif-1)-mediated autophagy [11,20]. In the present study, we found that both CDDO-Me and SN50 ameliorated aberrant mitochondrial hyperfusion by recovering NF-κB-PDI-mediated S-nitrosylation of DRP1. Interestingly, CK2 is an upstream regulator of both NF-κB S529 phosphorylation and AKT activation during clasmatodendrosis [11,39,40,41]. Considering oxidative stress-induced CK2 activation [11] and the anti-oxidant properties of CDDO-Me [24,25], our findings suggest that oxidative stress may cause CK2 hyperactivation eliciting clasmatodendrosis through NF-κB-PDI- and AKT-Bif-1-mediated signaling pathways.”

  1. The authors should rephrase the paper to indicate the effect of CDDO-Me and SN50. They should describe the effect of each agent separately. 

Answer: With respect to reviewer’s comments, we have also discussed the effect of each agent separately. 

I will be grateful if the manuscript could be reviewed and considered for publication.

Best Regards,

Ji-Eun Kim, Ph.D

Department of Anatomy and Neurobiology

Hallym University, College of Medicine

Chunchon, 24252, Kangwon-Do

Republic of Korea

Round 2

Reviewer 1 Report

The authors did not properly addressed the reviewer's concerns and the manuscript is still undermined by several concerns.

Below some examples.

Experimental evidence showing that clasmatodendrosis can be considered an autophagic cell death still remains limited. Additionally, research evidence demonstrated that, at least in specific physiopathological conditions, clasmatodendrosis-induced death is not dependent on autophagy. Thus, clasmatodendrosis cannot be called "autophagic cell death" per se.

Despite this notion, the authors continue to stress that autophagy is determinant in clasmatodendrosis, however the methodological approach used in this work to assess autophagy is extremely inadequate. LC3 immunofluorescence shown in this study is awful. To start, an increase in LC3 alone does not necessarily indicate increased autophagy, as it may also suggest blockade of autophagosomal clearance. When autophagy is induced, LC3I is converted in LC3II, that binds to the autophagosomal membrane. Thus, immunostaining becomes peculiar, with the appearance of several spots corresponding to LC3-immunodecorated autophagosomes. In this case, LC3 staining is diffuse and typical LC3 dots cannot be observed. The authors specifically indicated LC3 immunofluorescence as "LC3II": it is very weird, since the antibody used is not able to discriminate LC3II from LC3I forms. No experiments on autophagy flux were conducted on clasmatodendrosis.

The authors stated that "LAMP1 expression level is very low and undetectable in astrocytes in vivo". However, this observation is in contrast with published literature data showing clear and appreciable LAMP1 staining in in vivo astrocytes.

The considerations about the use of S.E.M., instead of S.D. in the statistical analysis, do not make sense. It is not correct that SEM directly represent statistical significance. SEM gives an idea of the accuracy of the mean, whereas SD represents the variability of the single observations.

Author Response

Dear, Reviewer 1;

I am enclosing herewith the revised version of our manuscript entitled CDDO-Me abrogates aberrant mitochondrial elongation in au-tophagic astroglial death (clasmatodendrosis) by regulating NF-κB-PDI-mediated S-nitrosylation of DRP1 (Ms. No.: ijms-2217732)”.

We appreciate the reviewer’s recommendations to improve our manuscript.

With respect to reviewers’ comments and suggestions, we addressed the reviewers' points.

My responses to comments are as followed;

Reviewer 1

  1. Experimental evidence showing that clasmatodendrosis can be considered an autophagic cell death still remains limited. Additionally, research evidence demonstrated that, at least in specific physiopathological conditions, clasmatodendrosis-induced death is not dependent on autophagy. Thus, clasmatodendrosis cannot be called "autophagic cell death" per se.

Answer: With respect to reviewer’s comments, we deleted “autophagic cell death” in the title, toned down that clasmatodendrosis is relevant to excessive autophagic process as followed;

“In previous studies, we had reported that clasmatodendrosis is Tdt-mediated dUTP Nick-End Labelling (TUNEL)-negative coagulative necrosis in astrocytes [3,9]. Later, we had found that vacuoles in clasmatodendritic astrocytes contain LC3 and LAMP1 signals. Since LC3 is required for the autophagosome formation and LAMP1 is a marker for lysosomal biogenesis, amounts and morphology [69,70,71,72], we reported that large vacuoles in clasmatodendritic astrocytes are autolysosomes [12,13]. LAMP1 is the predominant lysosomal membrane protein to maintain the integrity of the lysosomal membrane and the clearance of autophagosomes [69,70]. Cytoplasmic form LC3 (LC3-I) is diffusely observed in cell bodies, which is modified to LC3-II and concentrated in autophagosomes that exhibits granular puncta and a different mobility in electrophoresis [14,37,71,72]. Thus, an increase in LC3-II/LC3-I ratio is an indicative of activation of autophagy process [37,71,72]. In the present study, clasmatodendritic CA1 astrocytes showed the increased LAMP1 and LC3 expression, which were ameliorated by CDDO-Me and SN50. Furthermore, LC3-positive puncta were apparently detected in vacuolized astrocytes of vehicle-treated epilepsy rats. Western blot data also revealed that LAMP1, LC3-I and LC3-II densities were elevated in the epileptic hippocampus concomitant with the increased LC3-II/LC3-I ratio, which were attenuated by CDDO-Me and SN50. Compatible with the present data, Sakai et al. [71] report that clasmatodendrosis is relevant to UPS-mediated autophagy. Qin et al. [72] also demonstrate that ischemia-injured astrocytes contain numerous multimembrane vesicles described as typical for autophagosomes, which eventually fused with lysosomes in the cytoplasm, indicating that the autophagic/lysosomal pathway activation contributes to the decreased viability of astrocytes. Therefore, our findings suggest that excessive autophagy is involved in the pathogenesis of clasmatodendritic degeneration in the epileptic hippocampus, although it is unclear whether aberrant autophagy is the main cause of clasmatodendrosis or directly leads to astroglial degeneration.”

  1. Despite this notion, the authors continue to stress that autophagy is determinant in clasmatodendrosis, however the methodological approach used in this work to assess autophagy is extremely inadequate. LC3 immunofluorescence shown in this study is awful. To start, an increase in LC3 alone does not necessarily indicate increased autophagy, as it may also suggest blockade of autophagosomal clearance. When autophagy is induced, LC3I is converted in LC3II, that binds to the autophagosomal membrane. Thus, immunostaining becomes peculiar, with the appearance of several spots corresponding to LC3-immunodecorated autophagosomes. In this case, LC3 staining is diffuse and typical LC3 dots cannot be observed. The authors specifically indicated LC3 immunofluorescence as "LC3II": it is very weird, since the antibody used is not able to discriminate LC3II from LC3I forms. No experiments on autophagy flux were conducted on clasmatodendrosis.

Answer: LC3-II is typo-error in Figure 1. In addition, we had focused on the vacuoles and LC3 upregulation in clasmatodendritic astrocytes in Figure 1. With respect to reviewer’s comment, we inserted the high magnification photos showing LC3-positive dots in vacuolized astrocytes of chronic epilepsy rats as Figure 2A. We also provided data concerning the increased autophagy flux by applying Western blot for LC3 and LAMP1 in Figure 2B-F.

  1. The authors stated that "LAMP1 expression level is very low and undetectable in astrocytes in vivo". However, this observation is in contrast with published literature data showing clear and appreciable LAMP1 staining in in vivo astrocytes.

Answer: We meant that LAMP1 staining was very weak and undetectable in control astrocytes under low magnification photos. With respect to reviewer’s comments, we inserted the high magnification photos showing small LAMP1 positive dots in control astrocytes as insertions on Figure 1.

  1. The considerations about the use of S.E.M., instead of S.D. in the statistical analysis, do not make sense. It is not correct that SEM directly represent statistical significance. SEM gives an idea of the accuracy of the mean, whereas SD represents the variability of the single observations.

Answer: We meant that SEM bars on graph can be used to get easily a sense for whether or not a difference is significant. For example, when SEM bars overlap quite a bit, it's a clue that the difference is not statistically significant. When SEM bars do not overlap, it's a clue that the difference may be significant. With respect to reviewer’s comments, however, we used S.D. in the statistical analysis.  

I will be grateful if the manuscript could be reviewed and considered for publication.

Best Regards,

Ji-Eun Kim, Ph.D

Department of Anatomy and Neurobiology

Hallym University, College of Medicine

Chunchon, 24252, Kangwon-Do

Republic of Korea

Reviewer 2 Report

no comments

Author Response

We appreciate the help of reviewer to improve our manuscripts.

Reviewer 3 Report

International Journal of Molecular Sciences (Manuscript ID: ijms-2217732), Comments to the Authors:

Title: CDDO-Me abrogates aberrant mitochondrial elongation in au-tophagic astroglial death (clasmatodendrosis) by regulating NF-κB-PDI-mediated S-nitrosylation of DRP1

Comments

After reading the authors’ response to my comments, I believe the paper can be accepted for publication. 

Author Response

(The authors gave the same response as above.)

Round 3

Reviewer 1 Report

The authors properly addressed the major concerns raised by the reviewer. The manuscript now appears more balanced and the experimental plan, as well as the visualization of the results, is more convincing.

I recommend to change the title of Figure 2, that should be: "Effects of CDDO-Me and SN50 on autophagy process in the epileptic hippocampus". In other words, the term "flux" should be removed since the proposed experiments are conducted without an inhibitor of autophagosome-lysosome fusion (i.e. chloroquine or bafilomycin A), that is essential to perform flux experiments.

Author Response

Dear, Reviewer 1;

I am enclosing herewith the revised version of our manuscript entitled CDDO-Me abrogates aberrant mitochondrial elongation in au-tophagic astroglial death (clasmatodendrosis) by regulating NF-κB-PDI-mediated S-nitrosylation of DRP1 (Ms. No.: ijms-2217732)”.

We appreciate the reviewer’s recommendations to improve our manuscript.

With respect to reviewers’ comments and suggestions, we addressed the reviewers' points.

My responses to comments are as followed;

Reviewer 1

  1. The authors properly addressed the major concerns raised by the reviewer. The manuscript now appears more balanced and the experimental plan, as well as the visualization of the results, is more convincing.

I recommend to change the title of Figure 2, that should be: "Effects of CDDO-Me and SN50 on autophagy process in the epileptic hippocampus". In other words, the term "flux" should be removed since the proposed experiments are conducted without an inhibitor of autophagosome-lysosome fusion (i.e. chloroquine or bafilomycin A), that is essential to perform flux experiments.

Answer: With respect to reviewer’s comments, we corrected it.

I will be grateful if the manuscript could be reviewed and considered for publication.

Best Regards,

Ji-Eun Kim, Ph.D

Department of Anatomy and Neurobiology

Hallym University, College of Medicine

Chunchon, 24252, Kangwon-Do

Republic of Korea
